# Fault Diagnosis and Prediction System for Metal Wire Feeding Additive Manufacturing

**DOI:** 10.3390/s24134277

**Published:** 2024-07-01

**Authors:** Meng Xie, Zhuoyong Shi, Xixi Yue, Moyan Ding, Yujiang Qiu, Yetao Jia, Bobo Li, Nan Li

**Affiliations:** 1School of Electrical and Information Engineering, Xi’an Jiaotong University City College, Xi’an 710018, China; 2School of Electronics and Information, Northwestern Polytechnical University, Xi’an 710129, China; jiayetao@mail.nwpu.edu.cn

**Keywords:** additive manufacturing, fault diagnosis, LSTM neural network, LabVIEW

## Abstract

In the process of metal wire and additive manufacturing, due to changes in temperature, humidity, current, voltage, and other parameters, as well as the failure of machinery and equipment, a failure may occur in the manufacturing process that seriously affects the current situation of production efficiency and product quality. Based on the demand for monitoring of the key impact parameters of additive manufacturing, this paper develops a parameter monitoring and prediction system for the additive manufacturing feeding process to provide a basis for future fault diagnosis. The fault diagnosis and prediction system for metal wire supply and additive manufacturing utilizes STM 32 as its core, enabling the capture and transmission of temperature, humidity, current, and voltage data. The upper computer system, designed on the LabVIEW 2019 virtual instrument platform, incorporates an LSTM neural network model and facilitates a connection between LabVIEW and MATLAB 2019 to achieve the prediction function. The monitoring and prediction system established in this study is intended to provide basic research assistance in the field of fault diagnosis.

## 1. Introduction

Additive manufacturing, an advanced production technique, is extensively employed in fabricating prototypes, tooling, and fully functional products, as referenced in [1]. This technology has gained widespread application in various industries, including aerospace, automobile, aircraft, and medical fields, as documented in [2]. It offers numerous advantages and potential uses, such as minimizing material consumption, facilitating the production of complex and optimized structural designs, reducing waste, enabling remote production, allowing mixed materials and hybrid construction, and facilitating strengthening and repairs. However, additive manufacturing faces several challenges, barriers, and unanswered questions. These include, but are not limited to, the understanding of fundamental mechanical properties, ensuring the consistency of both short-term and long-term structural integrity, maintaining geometric accuracy and minimizing variability, improving the speed and cost-efficiency of the construction process, and addressing issues related to quality assurance, design, and certification, as discussed in [3]. Among these challenges, the quality and safety concerns surrounding metal additive manufacturing are particularly pressing and need to be urgently addressed, as highlighted in [4]. Therefore, there is a critical need to develop a fault diagnosis and prediction system specifically for metal wire supply in additive manufacturing. Such a system would intelligently monitor the metal wire additive manufacturing equipment, aiming to significantly reduce the system’s failure rate.

At present, there are still few studies on fault diagnosis and prediction in metal wire feeding additive manufacturing; the few studies mainly focus on the control of metal material composition, temperature, speed, and other parameters. For fault diagnosis and prediction, the main method is feature extraction [5]. Physics-based methods and data-driven methods are mainly adopted in feature extraction. Feng et al. [6] comprehensively reviewed the utilization of over 20 time–frequency analysis techniques in the diagnosis of mechanical equipment faults. These methods encompassed high-order spectrum analyses, both linear and nonlinear approaches, as well as techniques involving adaptive parameterization or no parameterization. Lei et al. [7] successfully extracted high-quality features from both time and frequency domains through a combination of empirical mode decomposition (EMD) and Hilbert demodulation. Tran et al. [8] significantly improved the fault identification rate in the diagnosis of reciprocating compressor valves by integrating the Deep Belief Network (DBN) with the Teager–Kaiser energy operation (TKEO) algorithm. Shao et al. [9], on the other hand, merged particle swarm optimization (PSO) with DBN, achieving superior identification accuracy, especially in scenarios where prior fault information was not available for rotary bearings. In addition, the fault diagnosis and prediction methods also include physical-based methods [10], data-driven methods [11], and hybrid methods.

LSTM (Long Short-Term Memory), a specialized recurrent neural network, effectively addresses the long-term dependency issues encountered in standard recurrent networks, making it suitable for equipment fault diagnosis and prediction. Wang et al. [12] introduced an LSTM-based signal prediction model tailored for FOCT (fiber optic current transformer). This model utilizes historical data to facilitate pattern recognition and system fault diagnosis. Xiang et al. [13] presented a novel method for wind turbine fault detection, integrating a convolutional neural network (CNN) with an LSTM, enhanced by an attention mechanism (AM). This model provides early warnings for anomalous states and identifies faulty components through prediction residuals. Binu et al. [14] emphasized analog circuit fault prediction utilizing LSTM to create an effective predictive model for anticipating faults before their occurrence. She et al. [15] combined CNN, LSTM, and Convolutional LSTM (Conv LSTM) for fault diagnosis and post-accident prediction of Loss of Coolant Accidents (LOCAs) in Nuclear Power Plants (NPPs). Zhang et al. [16] innovated a deep learning model, T2V-LSTM, designed for multi-level fault predictions, aiming to detect faults preemptively.

During fault diagnosis and prediction, equipment monitoring is essential to facilitate timely manual adjustments. LabVIEW, a widely accepted graphical programming language environment in industry, academia, and laboratories, serves as a standard tool for data acquisition and instrument control. Its applications span various monitoring scenarios, including college student physical monitoring [17], electrocardiogram technology [18], bench-top air-to-water heat pumps [19], and high-power femtosecond laser system stabilization [20].

Furthermore, data acquisition and retrieval heavily rely on the support of the underlying hardware system. A range of sensors are tasked with gathering pertinent data, while the microcontroller manages data processing and transmission. Shi et al. [21] employed sensors and a single-chip microcomputer to capture the motion data of table tennis players, thereby establishing an evaluation and scoring system for the players. Moreover, they utilized sensors to record flight data from Unmanned Aerial Vehicles (UAVs) and implemented data fusion techniques to predict the UAV’s flight status [22] and trajectory [5,23].

In this study, we present a critical parameter monitoring and prediction system for assisted fault diagnosis, specifically for wire feed additive manufacturing. This system integrates the STM32 microcontroller and the LabVIEW virtual instrument platform, aiming to enhance the reliability and performance of the manufacturing process. The core objective of this research is to construct a robust fault diagnosis and prediction system for metal wire feeding additive manufacturing. To achieve this, we have designed a comprehensive system that consists of two main components: the host computer system and the lower computer system. For the host computer system, we utilize the Long Short-Term Memory (LSTM) neural network model, integrated with joint debugging of LabVIEW and MATLAB. This combination enables the realization of an accurate prediction function, which is crucial for anticipating and preventing potential faults in the manufacturing process. The lower computer system, on the other hand, employs the STM32 microcontroller as its core. This microcontroller facilitates the acquisition and transmission of critical data, such as temperature, humidity, current, and voltage. The precision of these measurements is paramount for ensuring accurate monitoring and control of the manufacturing environment. Specifically, our system achieves a temperature accuracy of ±0.5 °C, a humidity accuracy of ±2%, a current measurement accuracy of ±0.1 A, and a voltage measurement accuracy of ±1 V. By combining these two systems, we aim to create a comprehensive fault diagnosis and prediction system for metal wire feeding additive manufacturing, thus enhancing the overall quality, reliability, and efficiency of the process.

## 2. System Scheme Design

### 2.1. Frame of Thought

Figure 1 depicts the construction idea of the metal wire feeding additive manufacturing fault diagnosis and prediction system proposed in this paper, which mainly includes the theoretical layer, the hardware layer, and the software layer.

#### 2.1.1. Theoretical Layer

The theoretical layer constructs a theoretical knowledge framework, which mainly includes the development of the additive manufacturing process and the fault principle of metal wire feeding, deep learning theoretical knowledge, the LSTM neural network model, and other prediction models.

#### 2.1.2. Hardware Layer

The hardware layer completes the design of the hardware circuit, builds the hardware circuit by selecting the appropriate components, realizes the collection of data, and sets the serial port configuration of the lower computer to connect with the upper computer, such that the data can be successfully transmitted to the upper computer.

#### 2.1.3. Software Layer

The software layer divides the system function modules and builds the fault diagnosis subsystem and the fault prediction subsystem. By receiving the data transmitted by the hardware layer and adjusting the parameter range of the front panel of the host computer, the system program control of the rear panel determines whether they are fault data, and the indicator light changes depending on whether the data are faulty. In the fault prediction part, the LSTM neural network prediction model is built, and LabVIEW and MATLAB are jointly debugged to train the model on the historical data time series set to realize the prediction of the data.

### 2.2. System Block Diagram

The system block diagram is shown in Figure 2. Using an STM32 single-chip microcomputer (China) as the core, the fault detection and prediction first collect the data and clarify the parameter range. Then, the single-chip microcomputer system analyzes the collected data, judges whether the data are within the normal range, converts the data within the parameter range into a text signal of “normal” or “fault”, transmits the data to the virtual reality panel of the host computer, collects data multiple times for each transmission, and realizes the fault diagnosis and prediction of the system hardware design.

The system is designed using the LabVIEW platform to realize the functions of data acquisition, data processing, LSTM neural network model construction, fault diagnosis, and prediction. The data acquisition module is mainly implemented using the sensor module in LabVIEW, which can collect various parameter data in the metal wire feeding additive manufacturing process in real time. The data processing module is mainly implemented through the data processing module in LabVIEW. The LSTM neural network model’s building blocks are mainly implemented using the neural network modules in LabVIEW, including the building, training, and testing of LSTM neural network models. The fault diagnosis and prediction module is mainly implemented through the human–machine interface module in LabVIEW, which can display the diagnosis and prediction results in a graphical way.

## 3. Establishment of Prediction Mode

### 3.1. Principle of Fault Diagnosis

Metal wire feed additive manufacturing is an advanced manufacturing process that enables the manufacturing of three-dimensional metal parts by layering metal materials layer by layer. In this process, the metal wire feeding system is an important part of the entire process chain. However, in practice, metal wire feeding systems can also fail, causing disruption to the manufacturing process and deterioration of part quality. Therefore, it is necessary to diagnose and deal with faults in metal wire feed additive manufacturing in a timely manner.

The basic principle of fault diagnosis in metal wire feeding additive manufacturing is to judge whether the system is working normally by monitoring and analyzing the parameters and status of each component in the metal wire feeding system so as to find out the cause of the failure. Specifically, the diagnostic process mainly includes data collection, data analysis, fault location, and treatment solutions. Data acquisition comprises obtaining various data about the metal wire feeding system through sensors and other devices, including voltage, current, speed, temperature, and other parameters, and recording and storing the data. Data analysis analyzes the recorded data, such as monitoring whether the individual components in the system are working properly and whether there are abnormal changes, such as sharp temperature rises, voltage anomalies, etc. Based on the results of the data analysis, the parts that may have problems are identified, and staff are guided to investigate the root cause of the failure. Finally, based on the results of fault location, the corresponding treatment plan is formulated, such as replacing parts, adjusting parameters, etc., to ensure the normal operation of the metal wire feeding system.

### 3.2. Failure Prediction Principle

The LSTM neural network model is the core part of the system, which is mainly used to establish fault diagnosis and prediction models for metal wire feed additive manufacturing. This design uses a prediction toolbox in MATLAB, which encapsulates the LSTM network in a similar toolbox-like form for ease of use. The MATLAB Forecasting Toolbox is a powerful toolbox in MATLAB for time series data analysis and forecasting. It provides many commonly used time series analysis methods, such as autoregressive (AR), moving average (MA), autoregressive moving average (ARMA), autoregressive integral moving average (ARIMA), and other models.

The forecasting toolbox uses classical time series analysis methods for data forecasting. Time series are treated as stochastic processes that build statistical models based on historical data and use that model to make predictions about future data. The toolbox removes noise and non-randomness from the data by inputting raw time series data into MATLAB for preprocessing operations, such as smoothing, detrending, differential, etc. Unknown parameters can be estimated using certain methods, such as maximum likelihood or least squares. Through residual analysis, the model goodness test, and other methods, the established model is tested to see if it conforms to the data. Established models are used to make predictions about future data.

### 3.3. Establishing the LSTM Neural Network Prediction Model 

A recurrent neural network (RNN) represents a neural network architecture specifically designed for processing sequential data, such as time series. As one of the early deep learning algorithms, RNNs exhibit adaptability, enabling them to integrate multiple preceding data points and analyze correlations within the input signal sequence. This capability enhances the accuracy of time series forecasting. When compared to traditional neural networks, RNNs excel in capturing the intricacies and patterns inherent in time series data.

However, standard RNNs face challenges in maintaining long-term dependencies due to various issues, such as vanishing or exploding gradients. This limitation paved the way for the introduction of Long Short-Term Memory (LSTM) networks. LSTM, a special type of RNN, mitigates these problems through its unique memory cell design.

The core of LSTM’s operation lies in its cell state, which acts as a conveyor belt running through the network. This design allows for minimal linear interactions, significantly reducing information loss during transmission. LSTM modulates the information flow to the cell state via a “gate” mechanism. Gates selectively permit information to pass through, enhancing the network’s ability to retain long-term dependencies. This innovative approach, schematically represented in Figure 3, underscores LSTM’s superiority in handling complex time series data compared to traditional RNNs. We adopt the LSTM network due to its proficiency in capturing and maintaining long-term dependencies, crucial for accurate time series analysis and forecasting.

However, it also has certain drawbacks, as when the training data set grows or the time interval between the data becomes larger, the new data passed to the neural network may exceed the scope of the initial hidden layer, causing the network to “forget” valid information. This is because during training, the neural network is not able to adequately process long-term memory information. The LSTM adopts a gated output method, i.e., three gates (input gate, forgetting gate, and output gate) and two states (cell state and hidden state), which can learn long-term information by default at the beginning of the design so as to achieve the long memory of the data input to the neural network.

The forgetting gate is used to decide whether to keep the information or not, indicating how much it should forget the state at the previous point in time, as shown in Equation (1).
(1)$$f_{t}=\sigma \left(W_{f}\cdot \left[h_{t-1},X_{t}\right]+b_{f}\right)$$

Take the output of the previous $h_{t-1}$ stage and the input of this stage $X_{t}$ as inputs, and then create an alternate $C_{t}^{\sim }$ one and use tanh functions to control the size of the information that needs to be added. The specific formula of the input gate is shown in Equations (2) and (3).
(2)$$i_{t}=\sigma \left(W_{i}\cdot \left[h_{t-1},X_{t}\right]+b_{i}\right)$$
(3)$$C_{t}^{\sim }=\tanh\left(W_{c}\cdot \left[h_{t-1},X_{t}\right]+b_{c}\right)$$

Ct is fed into tanh, the function, which determines its final output part. Multiply this output by the output state of the current time step $O_{t}$ to obtain the final output. This process provides good control over the flow of information, ensuring that the model is able to correctly capture important feature information in the sequence data. The output gate formula is shown in Equation (4).
(4)$$O_{t}=\sigma \left(W_{o}\cdot \left[h_{t-1},X_{t}\right]+b_{o}\right)$$

## 4. System Software Design

The flow diagram of the host computer system is shown in Figure 4, which realizes the entire execution process of the virtual panel.

When the correct account and password are entered, click the login button, and the metal wire feeding additive manufacturing fault diagnosis and prediction system will be successfully logged in. If any one of the account passwords is entered incorrectly, the system will automatically pop up an error window, and if the system cannot successfully log in to the system, you need to continue to enter the correct account and password to log in.

After successfully logging into the system, the interface will automatically enter the data acquisition and display panel and the interface will automatically display the control and serial port initialization. At this time, you can start serial port communication and collect data, and the fault diagnosis and display of the collected data can be completed through program settings. After the data display and fault diagnosis are completed, you can click the fault prediction button to jump to the data prediction panel. If you do not click the button, the page will stay in the data collection and display panel, and you can repeat the data collection and fault diagnosis. If you click the fault prediction button, the system will automatically jump to the data prediction panel.

If you choose to import a time series set, click the Forecast button to start data prediction. The system will automatically pop up the MATLAB training progress window. Wait for the training progress to be completed. The prediction data waveform module on the page will display the current prediction waveform and fault indicator. Select Continue Prediction to repeat the steps of the current page. Return to the previous interface to click the “Data Collection” button and end to turn off the system’s operation.

## 5. System Front Panel Design

The upper computer display interface consists of a landing page, a data collection and display page, and a data prediction page.

### 5.1. Establishing the LSTM Neural Network Prediction Model 

The login page is shown in Figure 5, which is composed of the system name and the user login module. After the system starts to run, the user enters the correct account and password and clicks to log in. You can successfully log into the metal wire feeding additive manufacturing fault diagnosis and prediction system.

The login interface consists of the account and password file module, the system name, and the user login module. The user can set the account and password path according to the needs of the C disk and D disk. Click Run. After the user enters the correct account and password, click Login, and you can successfully log into the metal wire feeding additive manufacturing fault diagnosis and prediction system. If any of the account passwords are entered incorrectly, you will not be able to log into the system correctly. The default account is 19091027, and the password is 19091027. If you need to change the account, password, and file path, you can change them in the corresponding file path in the upper left corner.

### 5.2. Data Collection and Display Page

As shown in Figure 6, the data acquisition and display page is composed of a plurality of modules, mainly including a serial port configuration module, a parameter data waveform display module, a parameter-range-setting module, a data display and diagnosis module, and a “fault prediction” jump button module. Through the waveform display of the collected data, the digital display, the fault light display, etc., the process from data acquisition to data display is presented from multiple angles so that the data are more intuitive and clearly displayed on the panel, and the jump button can directly enter the next sub-page.

#### 5.2.1. Serial Port Module

The function of serial port communication is to connect the upper computer and the lower computer to realize serial port communication and data transmission, and the required port parameters can be correctly configured to achieve serial port communication, which is COM1 by default. The VISA resource name is the exact name of VISA, and it is used to specify the port type, i.e., the serial port name of the PC segment. The baud rate is a parameter used to indicate the transmission speed of a serial port, and there are usually several options depending on the user’s needs. In this app, the baud rate is set to 9600 bps by default. Data bits are used to specify the number of bits occupied in a set of data, and the common setting is 8 bits. In addition, the parity bit is also an optional parameter, which can be configured as no check, odd check, or even check, and the default configuration is no check. The stop bit is used to mark the stop of character data, which can be set to 1 bit, 1.5 bits, or 2 bits. In this app, the stop bit is set to 1 bit. The sampling interval represents the time difference between two samples, which is set to 10 here.

#### 5.2.2. Parameter-Range-Setting Module

The parameter-range-setting module is mainly composed of the upper and lower limits required by the four parameters, which can be changed according to different needs, and the parameter range can also be adjusted in real time according to the actual situation. The display accuracy is 3 digits. The temperature range is −10–50 °C, the humidity range is 0–100%RH, the current range is 0–30 A, and the voltage range is 0–36 V. Among them, temperature accuracy is ±0.5 °C, humidity accuracy is ±2%, current measurement accuracy is ±0.1 A, and voltage measurement accuracy is ±1 V.

#### 5.2.3. Parameter Data Waveform Display Module

The real-time waveform display includes the desired temperature, humidity, current, and voltage, each consisting of four graphs. After the serial port successfully transmits data, the real-time waveform of this parameter will be displayed on the page every time the data are transmitted to the host computer. The ordinate of the waveform represents the amplitude of the parameter, and the initial state of the four waveforms from top to bottom is −1 to 1 by default. The temperature unit is °C, the humidity unit is %RH, the current unit is A, and the voltage data are V. The amplitude changes as the data are entered, and the abscissa represents the time, with the initial state of 08:00:00 1 January, 08:00:00 30 December, 08:00:00 29 December, and 08:00:00 28 December. As the data are entered, the time changes accordingly to the current moment. Through the waveform chart, you can intuitively see the data changes of different parameters, which is convenient for the subsequent detection of fault problems.

#### 5.2.4. Data Display and Diagnosis Module

The data display diagnostic module consists of four parameter fault indicators. The original state of the indicator is green. When the system receives any temperature, humidity, current, voltage, and other data that are not within the parameter range, the corresponding indicator will jump from the initial state of “green” to “red”. When the received temperature, humidity, current, voltage, and other data are within the parameter range, the corresponding indicator remains “green”. Normal or fault data will also be displayed in the data frame, and the display accuracy is 3 digits. The temperature range is −10–50 °C, the humidity range is 0–100%RH, the current range is 0–30 A, and the voltage range is 0–36 V. Among them, temperature accuracy is ±0.5 °C, humidity accuracy is ±2%, current measurement accuracy is ±0.1 A, and voltage measurement accuracy is ±1 V.

#### 5.2.5. “Fault Prediction” Page Jumps to the Button Module

The module consists of a jump button control with the word “fault diagnosis”. When the function of the data collection and display page is completed, click the button to enter the data prediction interface.

### 5.3. Data Prediction Page

Figure 7 shows the data prediction page, which consists of a historical data recording module, a parameter data prediction graph module, a data prediction training set module, and a prediction alarm module. The collected data are imported into the training model, and the prediction toolbox in the joint debugging MATLAB can be used to predict the failure of the data.

#### 5.3.1. History Module

The history module mainly comprises importing an Excel 2019 sheet control to record the data of the data collection and display panel. When the received data are within the normal range, the system will automatically save the normal data value of the current time node, and when any of the four parameters are not within the normal range, the history will not save the data. The initial interface does not display the specific path. Users can change the historical data record path at the bottom of the table according to their needs. The default historical path is C:\Users\25741\Desktop\Fault Prediction 1.3\Fault Diagnosis and Prediction 1.3\History.xlsx. The module can save 5 types of parameters, such as time, temperature, humidity, current, and voltage, at one time. The temperature range is −10–50 °C, the humidity range is 0–100%RH, the current range is 0–30 A, and the voltage range is 0–36 V. Among them, time accuracy includes the year, month, day, and minute, temperature accuracy is ±0.5 °C, humidity accuracy is ±2%, current measurement accuracy is ±0.1 A, and voltage measurement accuracy is ±1 V.

#### 5.3.2. Data Prediction Training Set Module

The data processing and prediction module analyzes and extracts the data received on the serial port simulator, carries out data preprocessing, and realizes data display, fault diagnosis, historical data recording, and data display through the selection and effective connection of various controls. Fault prediction and other functions, in which the data prediction function needs to be implemented in conjunction with MATLAB, establish a MATLAB program window in the program panel and write the neural network prediction toolbox program in MATLAB, where data are the program input function and YPred is the function to realize the prediction. A program box can predict the data of one parameter, and four program boxes need to be designed in this design to predict the predicted data values of four parameters: temperature, humidity, current, and voltage. The specific program diagram is shown in Figure 8.

The MATLAB block diagram is connected to the program processing module through the processing and analysis of the data to determine whether the data are within the normal range. If they are in the normal range, the predicted alarm lamp will remain in the initial state of “green”. If they are outside of the normal range, the predicted alarm lamp will jump from the initial state of “green” to “red”. Correspondingly, the processing module corresponding to temperature, humidity, current, and voltage is connected to the corresponding prediction program module to realize the prediction and alarm function of the four parameters. The module diagram is shown in Figure 9.

#### 5.3.3. Parametric Data Prediction Graph Module

The parameter data prediction chart module consists of four charts corresponding to parameters, and each chart consists of an image display part and a prediction data display part. Each chart shows part of the abscissa to represent the time. The default amplitude of the initial state is −1 to 1, and the real-time time will be displayed when the data are predicted. The ordinate represents the amplitude of the parameter change, and the default amplitude of the initial state is −1 to 1. The specific number of the predicted change amplitude will be displayed when the data are predicted. The temperature range is −10–50 °C, the humidity range is 0–100%RH, the current range is 0–30 A, and the voltage range is 0–36 V. Among them, temperature accuracy is ±0.5 °C, humidity accuracy is ±2%, current measurement accuracy is ±0.1 A, and voltage measurement accuracy is ±1 V. When the data are successfully predicted, a prediction waveform is displayed in the chart section. The prediction data display part will be based on the LSTM neural network model, which predicts the value of 40 time nodes after the current time node of each parameter.

#### 5.3.4. Predictive Alarm Module

The prediction alarm module is mainly composed of four parameter indicators, and the main function is to display the data after predicting the data and displaying the fault judgment. When the predicted data of a parameter are not within the range of the parameter setting, the corresponding indicator will jump from “green” to “red”, which means that there are fault data outside of the normal range in the predicted data. When the color of the indicator does not change, it means that the predicted future data are within the normal range.

#### 5.3.5. “Data Collection” Page Jump Button Module

The module consists of a jump button control with the word “data collection”. When the function of the fault prediction page is completed, click the button to enter the data prediction interface.

## 6. System Testing and Result Analysis

### 6.1. Upper Computer System Fault Diagnosis Test

#### 6.1.1. Landing Page Testing

Open the LabVIEW system and import the password file in the user login module in the upper left corner of the file into the following path: C:\Users\25741\Desktop\Fault Prediction 1.3\Fault Diagnosis and Prediction 1.3\Password.txt. Import the account file into the path C:\Users\25741\Desktop\Fault Prediction 1.3\Fault Diagnosis and Prediction 1.3\Account.txt. Select Start Running Program, enter the login page of the metal wire feeding additive manufacturing fault diagnosis and prediction system, enter the default account, 19091027, and the default password, 123456, and click Login to successfully log into the system. The login page can operate normally, as shown in Figure 10.

#### 6.1.2. Data Display and Acquisition Panel Testing

After successfully logging into the system page, the system will automatically jump to the data collection and display interface and configure the parameters required by the serial port module on the left, in which the VISA resource name is set to the COM1 port, the baud rate is set to 9600, and the data bit is set to 8 bits. At the same time, the parity bit is set to None, the stop bit is set to 0, and the sampling interval is set to 10. Select the upper and lower limits of the temperature from −10 °C to 50 °C, the upper and lower limits of the humidity from 0%RH to 1%RH, the upper and lower limits of the current from 0 A to 30 A, and the upper and lower limits of the voltage from 0 V to 36 V.

(1)Normal data reception test

The lower computer collects data and transmits them to the upper computer to test the performance of system data reception and data storage. The 10 sets of normal temperature, humidity, current, and voltage data transmitted by the debugging serial port assistant are 30 °C, 0.6%RH, 14 A, 24 V, 20 °C, 0.7%RH, 17 A, 14 V, 30 °C, 0.2%RH, 17 A, 14 V, 20 °C, 0.2%RH, 18 A, 14 V, 20 °C, 0.4%RH, 17 A, 14 V, 10 °C, 0.9%RH, 16 A, 14 V, 10 °C, 0.4%RH, 29 A, 14 V, 22 °C, 0.3%RH, 29 A, 24 V, 20 °C, 0.4%RH, 19 A, and 27 V. The corresponding waveform changes and data diagnostic display are shown in Figure 11.

The module realizes the display of real-time data regarding the temperature, humidity, current, and voltage, and it can dynamically change with the transmission of data. At the same time, in the data display and module diagnostics functions, the final data presented are the latest set of data, i.e., the temperature is 3 °C, the humidity is 02%RH, the current is 12 A, and the voltage is 4 V. The transmitted data are within the normal range of parameters, so the indicator alarm state is not triggered, and the fault display and diagnosis module can display the data normally.

(2)Historical Data Recording Test

As shown in Figure 12, the historical data display page synchronously displays trouble-free data parameters, and the real-time data of the temperature, humidity, current, and voltage parameters are 20 °C, 0.4%RH, 17 A, 14 V, 10 °C, 0.9%RH, 16 A, 14 V, 10 °C, 0.4%RH, 29 A, 14 V, 22 °C, 0.3%RH, 29 A, and 24 V. The four sets of data displayed in the real-time time of 23 May 2023 21:52 are 20 °C, 0.4%RH, 19 A, 27 V, 20 °C, 0.3%RH, 20 A, 27 V, 13 °C, 0.3%RH, 20 A, 27 V, 13 °C, 0.3%RH, 20 A, and 23 V. The real-time time of 23 May 2023 21:53 shows four sets of data, which are 25 °C, 0.3%RH, 21 A, 23 V, 21 °C, 0.2%RH, 21 A, 23 V, 13 °C, 0.2%RH, 21 A, 23 V, 13 °C, 0.2%RH, 11 A, and 24 V. The four sets of data displayed at 23 May 2023 21:54 are 3 °C, 0.5%RH, 11 A, 5 V, 23 °C, 0.5%RH, 12 A, 15 V, 23 °C, 0.3%RH, 12 A, 25 V, 13 °C, 0.4%RH, 12 A, and 25 V. The four sets of data displayed at 23 May 2023 21:55 are 13 °C, 0.1%RH, 12 A, 14 V, 23 °C, 0.6%RH, 12 A, 14 V, 3 °C, 0.7%RH, 12 A, 14 V, 3 °C, 0.2%RH, 12 A, and 4 V. There are a total of 5 sets of data, and each set is 1 min, for a total of 20 sets of data. The historical data logging module is operational, and while the historical data module is running, the other modules of the data prediction panel remain in their initial state.

The data from the Excel table are shown in Table 1. The two significant digits are retained in the table by default. Open the Excel table in the file path, query the data records of the corresponding time, and find that the corresponding data records in the Excel table are consistent with the real-time data display. Each column is time, temperature/°C, humidity/%RH, current/A, and voltage/V. As can be seen from the figure below, the time is consistent with that in the historical data recording module accurate to the year/month/day/hour/minute, and a total of 20 sets of data from 23 May 2023 21:51 to 23 May 2023 21:55 are saved. This indicates that the history module is functioning properly.

(3)Fault Data Reception Test

To test the fault diagnosis function of the system, the temperature is set to 60 °C, the humidity is set to 1.6%RH, the current is set to 40 A, and the voltage is set to 40 V through the serial port simulator, and it is sent to the host computer, as shown in Figure 13. In the real-time data display, this set of data is accurately displayed as temperature 60 °C, humidity 1.6%RH, current 40 A, and voltage 40 V, and the waveform diagram module can also be displayed normally. If the transmitted temperature, humidity, current, and voltage data are outside of the normal range of parameters, the indicator alarm state is triggered, the four indicators are jumped from the initial state of “green” to “red”, and the fault display and diagnosis module can display the data normally.

### 6.2. Upper Computer System Data Prediction Test

#### 6.2.1. Parameter Training Test

The model training to be used for data prediction needs to use a large amount of data for training. Set a time series set text, import the set time series set in the data prediction path, click the data prediction button, and wait for a few seconds to automatically pop up the training progress of the LSTM neural network model based on the joint debugging state.

According to the program panel settings, each parameter will be selected for training in the data in the time series set, and each parameter will undergo 800 training iterations. The abscissa parameter of the first curve graph in the training progress is the number of iterations, and the ordinate parameter RMSE is the abbreviation of root mean square error, which is used to measure the error of the prediction model. The value range is 0 to positive infinity. The smaller the value, the smaller the prediction error of the model, while the stronger the value, the stronger the prediction ability of the model.

As shown in Figure 14, the second graph represents the loss curve, with the abscissa parameter being the number of iterations and the ordinate parameter being the amount of loss. As the number of training sessions increases, the prediction accuracy of the model for the real results also increases, so the gap between the prediction results and the real results gradually decreases. In training, the loss function is often used to measure the difference between the predicted and true results of the model, and the resulting loss curve reflects the trend of the loss function value as the number of training sessions changes. Generally speaking, with the increase in the number of training sessions, the model’s ability to fit the training data will gradually increase, and the value of the loss function will gradually decrease until it tends to be stable. During the training process, the loss curve can be observed to determine whether the model is convergent and whether there is overfitting or underfitting or other problems, which provides a basis for model tuning.

#### 6.2.2. Automatically End Iterative Training

Due to the influence of various factors, such as computer configuration and computer running speed, the training iteration speed of parameters, such as temperature, humidity, current, and voltage, will be affected, which will affect the completion time and accuracy of data prediction to a certain extent.

As shown in Figure 15, when the system is trained on a computer with a processor of 12th Gen Intel(R) Core(TM) i5-12500H (Intel, Santa Clara, CA, USA), the training progress of one of the parameters takes a total of 17 min and 49 s from automatic iteration to the end, and the training end state is the maximum number of rounds completed, while the start time is 23 May 2023. At 22:13:50, the number of iterations in each round is 1, the maximum number of iterations is 800, the RMSE curve is in a decreasing state, the peak value is about 1.25, the loss curve is in a decreasing state, and the peak is about 0.8.

#### 6.2.3. Manually End the Iterative Training

As can be seen from Figure 16, the time required to complete the model training is 17 min and 49 s, so, in order to save time, the MATLAB training progress window has a manual stop training iteration mode. When affected by the above factors, the training iteration speed will slow down, and you can choose whether to manually stop the training iteration according to your needs. When the training progress graph of a parameter starts running, the training progress graph is manually stopped at a random point in time. At this time, the training end state is a manual stop, the start time is 23 May 2023 22:31:43, the number of iterations in each round is 1, the maximum number of iterations is 800, the training is 800 rounds, it stops at the 325th round, the iteration is 800 times, it stops at the 325th time, and it lasts 11 min and 25 s. The RMSE curve showed a decreasing state with a peak value of about 1.03, and the loss curve showed a decreasing state with a peak value of about 0.58. Figure 15 shows the diagram of manually stopping the training iteration.

Although manually stopping the training iteration accelerates the model training speed, due to the insufficient number of iterations and insufficient training progress, the data error of the parameter prediction result is obvious compared with the parameter prediction result that is automatically stopped, but it does not affect the completion of the data prediction.

#### 6.2.4. Prediction Data Anomalies

When there is an abnormality in the prediction data, the abnormal parameters can be clearly observed from the prediction alarm module, and the number of significant digits in the data display can also be changed according to the demand. The significant digits are reserved for six digits in this prediction. As can be seen from the graph, the waveform of the temperature prediction data graph is significantly different from the waveform of other parameter data, and its peak value is far beyond the set range. The humidity varies roughly between 0%RH and 1%RH, the current varies between 0 A and 30 A, and the voltage varies between 0 V and 32 V. The temperature prediction alarm indicator at the prediction alarm module jumps from “green” to “red” in the initial state, indicating that the temperature parameter may have abnormal data in the predicted data of 40 nodes. This is consistent with the consequences that may result from manually stopping the temperature parameter training iteration before, as the specific prediction data below the waveform are displayed normally, and the indicator can display the prediction alarm information. The specific test diagram is shown in Figure 17.

#### 6.2.5. The Forecast Data Are Normal

When the training progress of the four parameters of temperature, humidity, current, and voltage is completed, close or minimize the four training progress pages, and the data waveform module will display the predicted data change waveform accordingly. The historical data saving interface will not be affected.

As shown in Figure 18, the predicted temperature data waveform shows that the temperature varies from 0.46 °C to 0.49 °C, the humidity varies from 0.2%RH to 0.7%RH, the current varies from 27.95 A to 28.15 A, and the voltage varies from 31.5 V to 34 V. The corresponding data display of each time node is located below the data prediction waveform chart, and there is an adjustment button on the far left of the data display. The prediction data can be queried in turn by clicking the up button, and up to 40 pieces of data can be queried. As can be seen from the figure, the predicted data are all normal data, so the alarm indicator of the prediction alarm module remains in the initial state of “green”.

## 7. Conclusions

In this paper, the design of a critical parameter monitoring and prediction system for fault diagnostic studies for additive manufacturing of metal wire feed is designed using STM32 as the core. The system collects, transmits, and predicts critical data, such as temperature, humidity, current, and voltage.

The host computer system is fashioned on the LabVIEW virtual instrument platform, integrating an LSTM neural network model. This integration, coupled with joint debugging between LabVIEW and MATLAB, facilitates advanced prediction capabilities. Our analysis of selected parameter data reveals that the LSTM neural network proves highly effective in diagnosing and predicting faults within the metal wire feeding additive manufacturing process.

The system boasts remarkable accuracy, with temperature readings within ±0.5 °C, humidity within ±2%, current measurements at ±0.1 A, and voltage within ±1 V. Its data acquisition range covers temperatures from −10 to 50 °C, humidity from 0% to 100%RH, currents from 0 to 30 A, and voltages from 0 to 36 V. By processing and analyzing the transmitted data, our system significantly enhances the intelligent monitoring, fault diagnosis, and prediction abilities of metal wire feeding additive manufacturing equipment. This not only reduces the system’s failure rate but also ensures excellent sensor scalability, paving the way for more efficient and reliable manufacturing processes.

At present, the metal wire feeding additive manufacturing fault diagnosis and prediction system established in this study mainly realizes the monitoring and prediction of parameters in the additive manufacturing process. In the future, we will focus on utilizing the monitored key parameters to achieve diagnosis of faults in the additive manufacturing process.

## Figures and Tables

**Figure 1 sensors-24-04277-f001:**
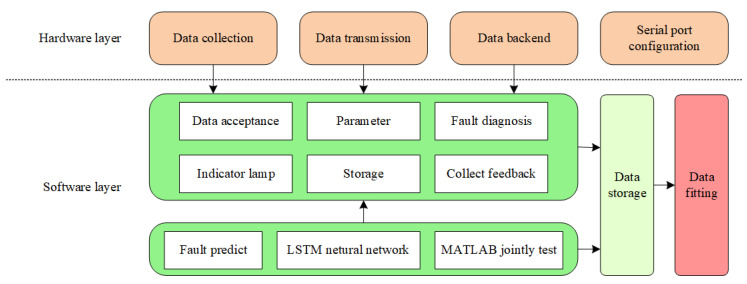
Framework diagram.

**Figure 2 sensors-24-04277-f002:**
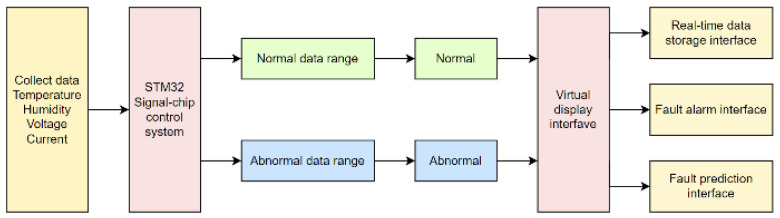
System block diagram.

**Figure 3 sensors-24-04277-f003:**
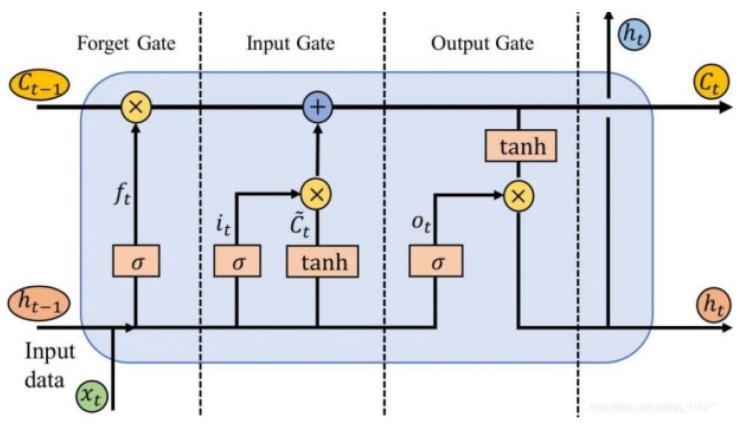
Schematic diagram of LSTM.

**Figure 4 sensors-24-04277-f004:**
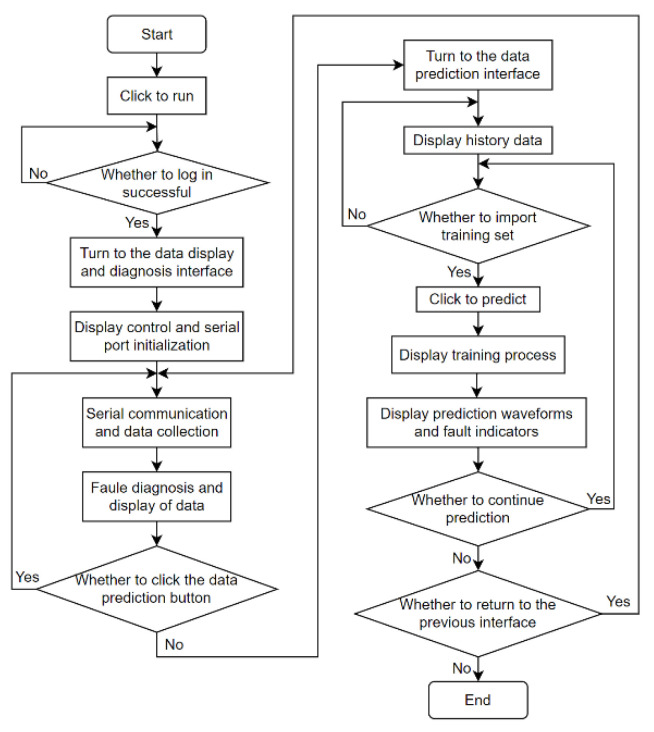
Flow chart of the host computer system.

**Figure 5 sensors-24-04277-f005:**
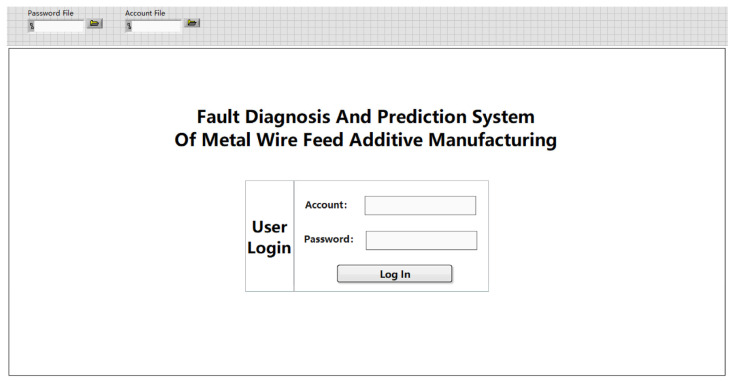
Landing page.

**Figure 6 sensors-24-04277-f006:**
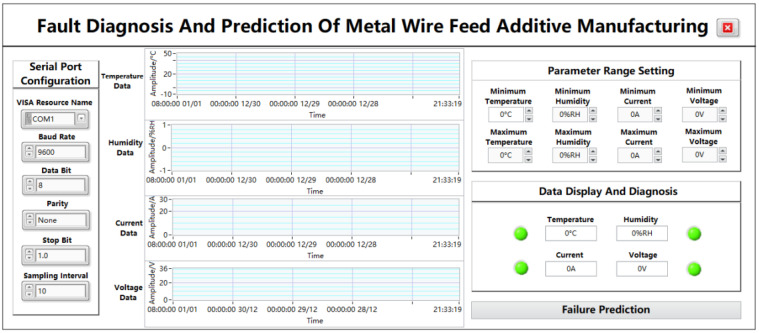
Data collection and display page.

**Figure 7 sensors-24-04277-f007:**
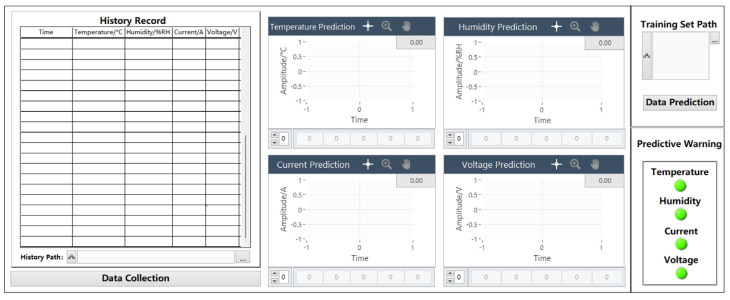
Data prediction panel.

**Figure 8 sensors-24-04277-f008:**
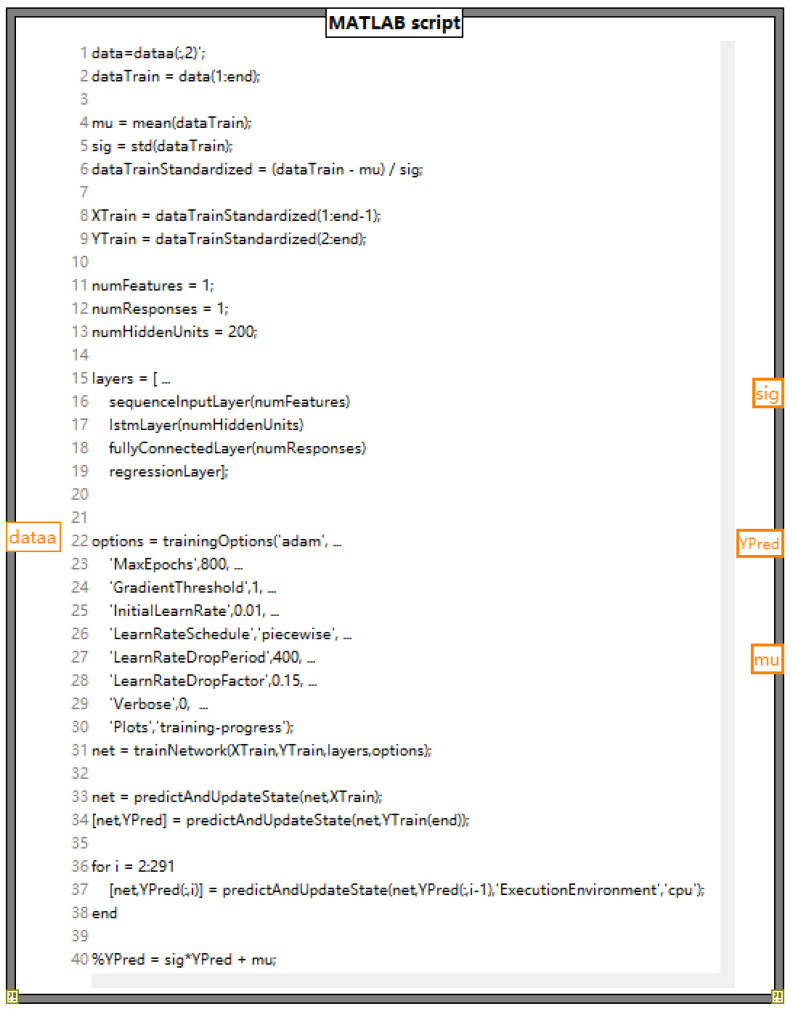
Prediction program diagram.

**Figure 9 sensors-24-04277-f009:**
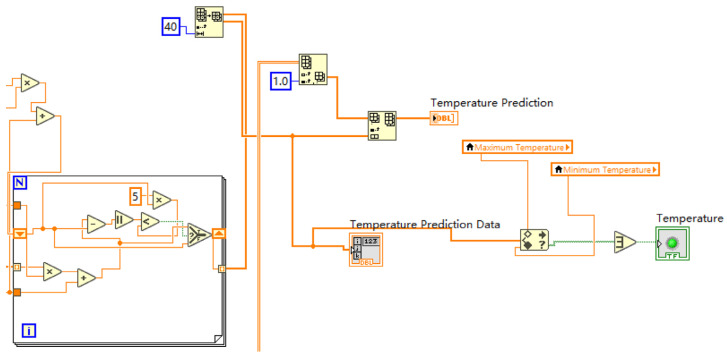
Diagram of the data prediction processor.

**Figure 10 sensors-24-04277-f010:**
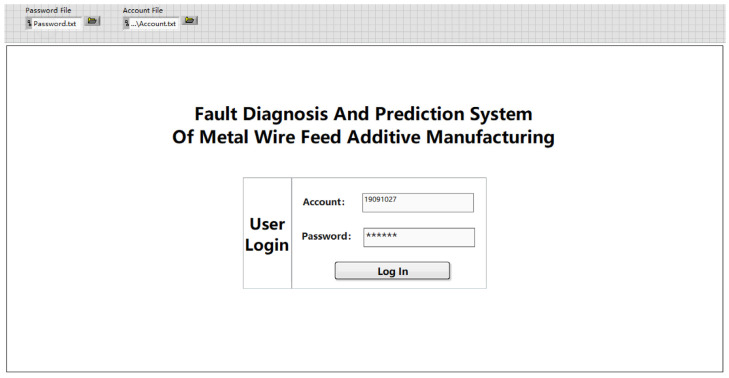
System login page.

**Figure 11 sensors-24-04277-f011:**
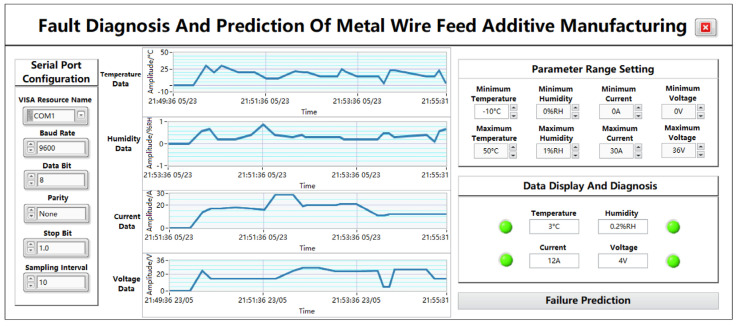
Real-time data graph displayed by the host computer.

**Figure 12 sensors-24-04277-f012:**
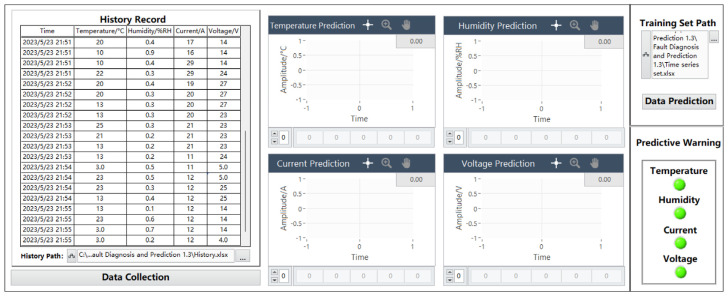
Historical data records are displayed.

**Figure 13 sensors-24-04277-f013:**
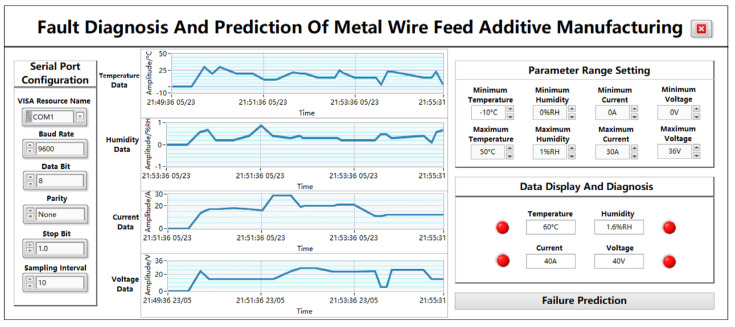
Fault data display chart.

**Figure 14 sensors-24-04277-f014:**
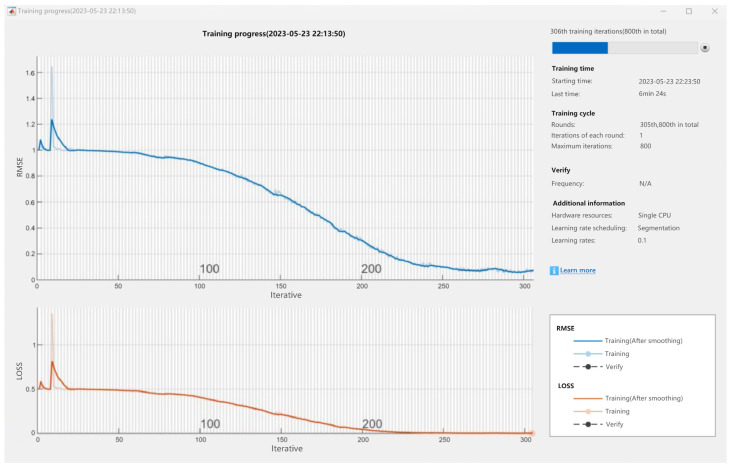
Iterative graph of parameter training.

**Figure 15 sensors-24-04277-f015:**
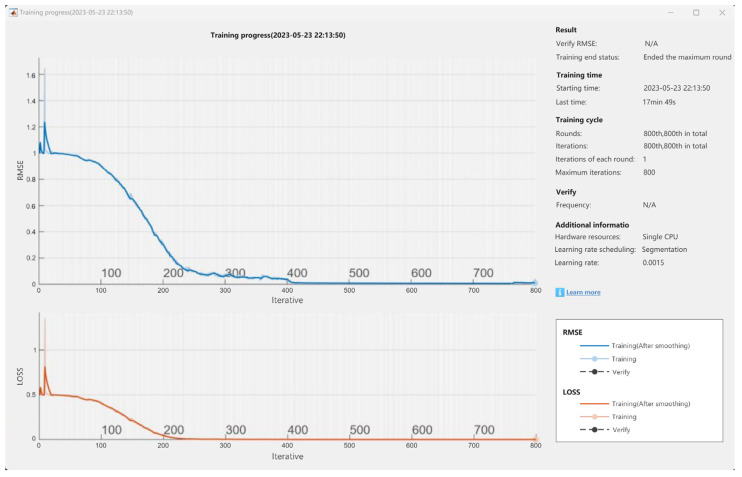
Auto-end iteration diagram.

**Figure 16 sensors-24-04277-f016:**
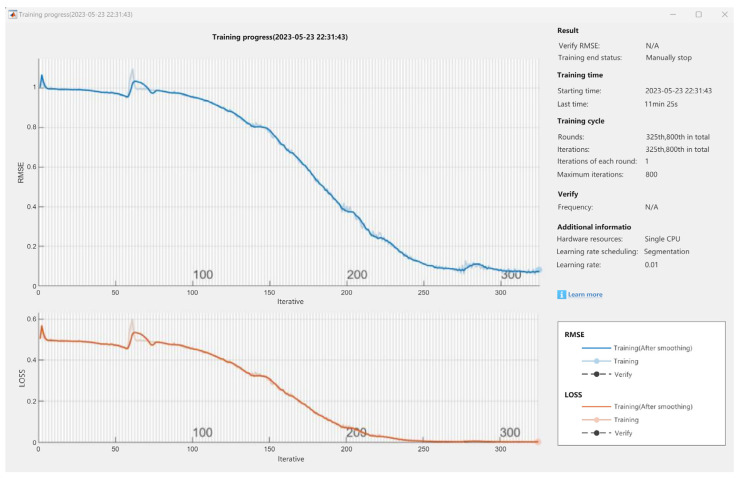
Manually stopping an iteration diagram.

**Figure 17 sensors-24-04277-f017:**
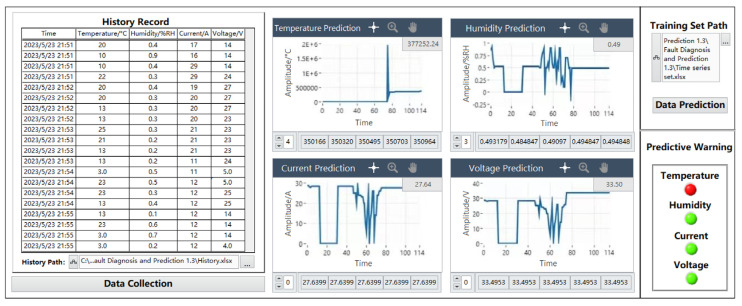
Abnormal Data prediction chart.

**Figure 18 sensors-24-04277-f018:**
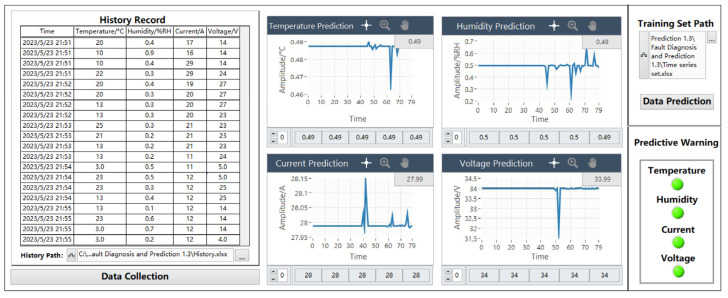
Normal Data prediction chart.

**Table 1 sensors-24-04277-t001:** Excel data graph.

Time	Temperature/°C	Humidity/%RH	Current/A	Voltage/V
23 May 2023 21:51	20	0.4	17	14
23 May 2023 21:51	10	0.9	16	14
23 May 2023 21:51	10	0.4	29	14
23 May 2023 21:51	22	0.3	29	24
23 May 2023 21:52	20	0.4	19	27
23 May 2023 21:52	20	0.3	20	27
23 May 2023 21:52	13	0.3	20	27
23 May 2023 21:52	13	0.3	20	23
23 May 2023 21:53	25	0.3	21	23
23 May 2023 21:53	21	0.2	21	23
23 May 2023 21:53	13	0.2	21	23
23 May 2023 21:53	13	0.2	11	24
23 May 2023 21:54	3.0	0.5	11	5.0
23 May 2023 21:54	23	0.5	11	5.0
23 May 2023 21:54	23	0.3	12	25
23 May 2023 21:54	13	0.4	12	25
23 May 2023 21:55	13	0.1	12	14
23 May 2023 21:55	23	0.6	12	14
23 May 2023 21:55	3.0	0.7	12	14
23 May 2023 21:55	3.0	0.2	12	4.0

## Data Availability

Data are contained within the article.

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
