# Peer review of "Fault Diagnosis and Prediction System for Metal Wire Feeding Additive Manufacturing"

_sensors, 2024, doi:10.3390/s24134277_

Round 1

Reviewer 1 Report

Comments and Suggestions for Authors

   1.The article only mentions that the failure of metal wire feeding additive manufacturing equipment can be determined by temperature, humidity, current, voltage and other parameters, but it does not analyze in detail the relationship between each type of parameter and the failure mechanism of the equipment.The mechanism of equipment failure is not analyzed in detail.

      3.The framework diagram of ideas does not correctly reflect the relationship between different layers, such as the relationship between the data layer and the hardware layer, and lacks the most important "decision layer".

      3. The ideas in the paper are not clear enough.Fault diagnosis and prediction should be discussed as two separate units in the paper, and the fault diagnosis unit should be placed before the prediction unit.

     4. The paper lacks the establishment of fault diagnosis model.

Comments on the Quality of English Language

 The translation is accurate and the wording is appropriate, but it is necessary to improve the application level of grammar.

Author Response

Dear Editors and Reviewers:

Thank you for your letter and for the reviewers’ comments concerning our manuscript entitled “Fault Diagnosis and Prediction System for Metal Wire Feeding Additive Manufacturing” (Manuscript Number: sensors-3070492). Those comments are all valuable and very helpful for revising and improving our paper, as well as the important guiding significance to our researches. We have studied comments carefully and have made correction which we hope meet with approval.

Revised portion are marked in yellow in the paper. The main corrections in the paper and the responds to the reviewers’ comments are as flowing:

Response to Reviewer #1:

[Comment 1] The article only mentions that the failure of metal wire feeding additive manufacturing equipment can be determined by temperature, humidity, current, voltage and other parameters, but it does not analyze in detail the relationship between each type of parameter and the failure mechanism of the equipment. The mechanism of equipment failure is not analyzed in detail.

Reply: We are very grateful to the reviewers for raising questions about fault diagnosis, however, this paper focuses on monitoring the key parameters in the metal wire feeding process to provide a basic research platform for research in the field of fault diagnosis by building a more accurate monitoring platform. Meanwhile, we have placed the study of fault diagnosis as a future research direction in the conclusion of the manuscript.

How the paper is modified:   The before and after revisions about conclusion of our latest manuscript submission are as follows:

Before modification

After modification

[Comment 2] The framework diagram of ideas does not correctly reflect the relationship between different layers, such as the relationship between the data layer and the hardware layer, and lacks the most important "decision layer".

Reply: We are very grateful to the reviewers for pointing out the framing problem of Figure 1, which has been corrected in our latest manuscript submission. Meanwhile, regarding the "decision layer", our manuscript mainly provides a detection system and prediction of the monitored parameters, which will serve the field of fault diagnosis, and this study does not deal with fault diagnosis methods, so there is no "decision layer" in Figure 1.

How the paper is modified: The before and after revisions about figure1 of our latest manuscript submission are as follows:

Before modification

After modification

[Comment 3] The ideas in the paper are not clear enough. Fault diagnosis and prediction should be discussed as two separate units in the paper, and the fault diagnosis unit should be placed before the prediction unit.

Reply: We are very grateful to the reviewers for suggesting a framework for the paper, however, our manuscript does not contain a description of the modelling approach for fault diagnosis. We focus on fault diagnosis by creating a monitoring system that monitors key parameters in the additive manufacturing process to aid in fault diagnosis. We have also revised the abstract section of the manuscript.

How the paper is modified: The before and after revisions about abstract of our latest manuscript submission are as follows:

Before modification

After modification

[Comment 4] The paper lacks the establishment of fault diagnosis model.

Reply: This paper focuses on monitoring the critical parameters of the wire additive manufacturing process and alerting the experimenters when there is a malfunction. We would also like to thank the reviewers for the innovative suggestion of "Fault Diagnostic Modelling", which will be further investigated in our future research. We have added directions for future research in the conclusion section of our latest manuscript submission.

How the paper is modified: The before and after revisions about conclusion of our latest manuscript submission are as follows:

After modification

At present, the metal wire feeding additive manufacturing fault diagnosis and prediction system established in this study mainly realises the monitoring and prediction of parameters in the additive manufacturing process. In the future, we will focus on utilising the monitored key parameters to achieve diagnosis of faults in the additive manufacturing process.

According to the reviewers’ comments, we have made extensive modifications to our manuscript and supplemented extra data to make our results convincing. Thank you again for your positive comments and valuable suggestions to improve the quality of our manuscript.

We appreciate for editors and reviewers’ warm work earnestly, and hope that the correction will meet with approval.

Yours sincerely,

Zhuoyong Shi

June 19, 2024

Reviewer 2 Report

Comments and Suggestions for Authors

My opinion is to reject this article

The paper presents a method for intelligent monitoring, fault diagnosis, and prediction of metal wire additive manufacturing equipment. However, I believe that it does not meet the standards for acceptance as an article. The overall architecture and description of experimental results resemble more of a project conclusion report.

2. The introduction of the article is disorganized and resembles a collection of articles with similar methods.

3. The article mentions the use of LSTM to solve the long-term dependence problem common in general recurrent neural networks. However, there should be an analysis on the necessity of using LSTM. Is the effect of general neural networks sufficient? Can other recurrent neural networks serve the same purpose?

4. The article discusses machine failure indicators but does not address fault type identification or whether it is necessary to identify fault types. When a fault indicator lights up, how are maintenance recommendations determined?

5.The full text simply describes work steps without reflecting innovation or comparing with other methods which are important aspects for demonstrating research work through comparative experiments."

Comments on the Quality of English Language

Need Improvement

Author Response

Dear Editors and Reviewers:

Thank you for your letter and for the reviewers’ comments concerning our manuscript entitled “Fault Diagnosis and Prediction System for Metal Wire Feeding Additive Manufacturing” (Manuscript Number: sensors-3070492). Those comments are all valuable and very helpful for revising and improving our paper, as well as the important guiding significance to our researches. We have studied comments carefully and have made correction which we hope meet with approval.

Revised portion are marked in yellow in the paper. The main corrections in the paper and the responds to the reviewers’ comments are as flowing:

Response to Reviewer #2:

[Comment 1] The paper presents a method for intelligent monitoring, fault diagnosis, and prediction of metal wire additive manufacturing equipment. However, I believe that it does not meet the standards for acceptance as an article. The overall architecture and description of experimental results resemble more of a project conclusion report.

Reply: Many thanks to the reviewers for raising the issue of the structure of the manuscript, we have refined the structure of the main elements of the manuscript in our latest submission, as well as highlighting the research framework of the manuscript in the abstract.

How the paper is modified:   The before and after revisions about Abstract of our latest manuscript submission are as follows:

Before modification

After modification

[Comment 2] The introduction of the article is disorganized and resembles a collection of articles with similar methods.

Reply: We have reorganized the literature review etc. in the introduction to increase the readability of the introductory section.

How the paper is modified: The before and after revisions about introduction of our latest manuscript submission are as follows:

1

Before modification

After modification

2

Before modification

After modification

3

Before modification

After modification

4

Before modification

After modification

5

Before modification

After modification

6

Before modification

After modification

[Comment 3] The article mentions the use of LSTM to solve the long-term dependence problem common in general recurrent neural networks. However, there should be an analysis on the necessity of using LSTM. Is the effect of general neural networks sufficient? Can other recurrent neural networks serve the same purpose?

Reply: We are very grateful to the reviewers for suggesting deficiencies regarding the description of the LSTM network. Since the key parameters of the additive manufacturing process for metal wires are mainly governed by time, a time-dependent LSTM network is used for optimisation.

How the paper is modified: In the description section 3.3 about LSTM we have improved it, before and after as follows:

Before modification

After modification

[Comment 4] The article discusses machine failure indicators but does not address fault type identification or whether it is necessary to identify fault types. When a fault indicator lights up, how are maintenance recommendations determined?

Reply: This paper focuses on monitoring the critical parameters of the metal wire additive manufacturing process and alerting the experimenters when there is a malfunction. We would also like to thank the reviewers for the innovative suggestion of "Establishing maintenance recommendations for machine failures", which will be further investigated in future research. We have added directions for future research to the conclusions in our latest manuscript submission.

How the paper is modified: A description of the improvements in our conclusion section follows:

After modification (future research)

 [Comment 5] The full text simply describes work steps without reflecting innovation or comparing with other methods which are important aspects for demonstrating research work through comparative experiments."

Reply: Many thanks to the reviewers for raising the issue of lack of clarity of the main contributions of this paper, which we have restated in the introduction to the manuscript. This study focuses on monitoring the key parameters (temperature, humidity, current, voltage, etc.) of the metal wire additive manufacturing process and building an engineering usable monitoring system. The accuracy of the key parameters of the monitoring process in this paper has also been restated in the conclusion section, which underpins the applied innovation of the study.

How the paper is modified: We have amended the main contributions in the conclusions section, before and after the modifications, as follows:

Before modification

After modification

According to the reviewers’ comments, we have made extensive modifications to our manuscript and supplemented extra data to make our results convincing. Thank you again for your positive comments and valuable suggestions to improve the quality of our manuscript.

We appreciate for editors and reviewers’ warm work earnestly, and hope that the correction will meet with approval.

Yours sincerely,

Zhuoyong Shi

June 19, 2024

Reviewer 3 Report

Comments and Suggestions for Authors

The paper develops a fault diagnosis and prediction system for metal wire feeding additive manufacturing using STM 32 for data acquisition and an LSTM neural network with LabVIEW and Matlab integration. This system enhances intelligent monitoring and reduces failure rates, improving production efficiency and product quality. Several major concerns and specific comments are listed below:

1.     The reviewed literature is mostly outdated. The author should focus on research outcomes from the past three years in this journal and related journals. Additionally, the citation format is incorrect. For example, citation 14 should follow the format: Binu et al. [14].

2.     The final paragraph of the introduction is confused with the conclusion. The author should clearly distinguish between the introduction and the conclusion.

3.     The expression in section 2.1 is imprecise and does not conform to the structure of neural network construction. If the author intends to convey the construction approach of the model, they should use flowcharts or pseudocode rather than a mind map.

4.     The author needs to use quantitative expressions to demonstrate the effectiveness of the proposed method and software.

Author Response

Dear Editors and Reviewers:

Thank you for your letter and for the reviewers’ comments concerning our manuscript entitled “Fault Diagnosis and Prediction System for Metal Wire Feeding Additive Manufacturing” (Manuscript Number: sensors-3070492). Those comments are all valuable and very helpful for revising and improving our paper, as well as the important guiding significance to our researches. We have studied comments carefully and have made correction which we hope meet with approval.

Revised portion are marked in yellow in the paper. The main corrections in the paper and the responds to the reviewers’ comments are as flowing:

Response to Reviewer #3:

[Comment 1] The reviewed literature is mostly outdated. The author should focus on research outcomes from the past three years in this journal and related journals. Additionally, the citation format is incorrect. For example, citation 14 should follow the format: Binu et al. [14].

Reply: Many thanks to the reviewers for raising the issue of references in our manuscript, and we have updated some of the references, as well as the citation of references, in our latest submission.

How the paper is modified:   The before and after revisions about introduction of our latest manuscript submission are as follows:

Before modification

After modification

[Comment 2] The final paragraph of the introduction is confused with the conclusion. The author should clearly distinguish between the introduction and the conclusion.

Reply: We are very grateful to the reviewers for pointing out the confusion between our introduction and conclusion statements. We have updated the introductory statement and the conclusion separately to distinguish the focus of the two sections. We have made corrections in the latest submission of the manuscript.

How the paper is modified: A description of the improvements in our introduction section follows:

Before modification

After modification

A description of the improvements in our conclusion section follows:

Before modification

After modification

[Comment 3] The expression in section 2.1 is imprecise and does not conform to the structure of neural network construction. If the author intends to convey the construction approach of the model, they should use flowcharts or pseudocode rather than a mind map.

Reply: Many thanks to the reviewers for their questions about the structure of the neural network in Section 2.1, in fact we wanted to emphasize the framework of the whole system rather than the structure of the neural network in Section 2.1. We have improved the textual content of the description in Section 2.1, and we have also redrawn Figure 1.

How the paper is modified: We have improved the description section in Section 2.1 and also redrawn Figure 1.

Before modification

After modification

[Comment 4] The author needs to use quantitative expressions to demonstrate the effectiveness of the proposed method and software.

Reply: We are very grateful to the reviewers for raising the issue of inaccuracies in the description of the section of our article in which the system was analysed and tested. We have made a statement about this issue in the conclusion section of our latest submitted manuscript.

How the paper is modified: A description of the improvements in our conclusion section follows:

Before modification

After modification

According to the reviewers’ comments, we have made extensive modifications to our manuscript and supplemented extra data to make our results convincing. Thank you again for your positive comments and valuable suggestions to improve the quality of our manuscript.

We appreciate for editors and reviewers’ warm work earnestly, and hope that the correction will meet with approval.

Yours sincerely,

Zhuoyong Shi

June 19, 2024

Round 2

Reviewer 1 Report

Comments and Suggestions for Authors

The clarity of the ideas and the rationality of the structure of the manuscript are important criteria for measuring the quality of the manuscript. According to common sense: only on the premise of diagnosing the existence of faults can the prediction of faults be carried out, so the relationship between fault diagnosis and fault prediction belongs to the step-by-step. Therefore, fault diagnosis belongs to the content that should be emphasized in the manuscript. We hope that you will carefully organize the ideas of your manuscript, check the problems in the structure of the manuscript, and revise it before resubmitting it.

Comments on the Quality of English Language

The quality of the English language is a prerequisite for a manuscript to reach the preliminary review stage. Although the English language quality of the manuscript has been greatly improved after the revision, there are still some grammatical errors, so please check it carefully and revise it.

Author Response

Dear Editors and Reviewers:

Thank you for your letter and for the reviewers’ comments concerning our manuscript entitled “Fault Diagnosis and Prediction System for Metal Wire Feeding Additive Manufacturing” (Manuscript Number: sensors-3070492). Those comments are all valuable and very helpful for revising and improving our paper, as well as the important guiding significance to our researches. We have studied comments carefully and have made correction which we hope meet with approval.

Revised portion are marked in yellow in the paper. The main corrections in the paper and the responds to the reviewers’ comments are as flowing:

Response to Reviewer #1:

[Comment 1] The clarity of the ideas and the rationality of the structure of the manuscript are important criteria for measuring the quality of the manuscript. According to common sense: only on the premise of diagnosing the existence of faults can the prediction of faults be carried out, so the relationship between fault diagnosis and fault prediction belongs to the step-by-step. Therefore, fault diagnosis belongs to the content that should be emphasized in the manuscript. We hope that you will carefully organize the ideas of your manuscript, check the problems in the structure of the manuscript, and revise it before resubmitting it.

Reply: We are very grateful to the reviewers for pointing out a possible problem in the structure of our manuscript: fault prediction without fault diagnosis. We are very sorry that the reviewers misunderstood us because of the lack of description of our terminology. We were hoping to help scholars in the field of fault diagnosis and prediction by predicting future values of the key parameters we have collected for such studies. We have changed these potentially confusing descriptions in the manuscript.

How the paper is modified:   The before and after revisions about Abstract of our latest manuscript submission are as follows:

Before modification

After modification

The before and after revisions about introduction of our latest manuscript submission are as follows:

Before modification

After modification

The before and after revisions about conclusion of our latest manuscript submission are as follows:

Before modification

After modification

[Comment 2] The quality of the English language is a prerequisite for a manuscript to reach the preliminary review stage. Although the English language quality of the manuscript has been greatly improved after the revision, there are still some grammatical errors, so please check it carefully and revise it.

Reply: We are very grateful to the reviewers for their strict demands on our English grammar. We have checked the article again for grammatical words.

According to the reviewers’ comments, we have made extensive modifications to our manuscript and supplemented extra data to make our results convincing. Thank you again for your positive comments and valuable suggestions to improve the quality of our manuscript.

We appreciate for editors and reviewers’ warm work earnestly, and hope that the correction will meet with approval.

Yours sincerely,

Zhuoyong Shi

June 25, 2024

Reviewer 2 Report

Comments and Suggestions for Authors

comment5 is not well addressed, and the authors sidestep my suggestion to "compare with other approaches to demonstrate the innovative nature of this paper."

Comments on the Quality of English Language

Good

Author Response

Dear Editors and Reviewers:

Thank you for your letter and for the reviewers’ comments concerning our manuscript entitled “Fault Diagnosis and Prediction System for Metal Wire Feeding Additive Manufacturing” (Manuscript Number: sensors-3070492). Those comments are all valuable and very helpful for revising and improving our paper, as well as the important guiding significance to our researches. We have studied comments carefully and have made correction which we hope meet with approval.

Revised portion are marked in yellow in the paper. The main corrections in the paper and the responds to the reviewers’ comments are as flowing:

Response to Reviewer #2:

[Comment 1] comment5 is not well addressed, and the authors sidestep my suggestion to "compare with other approaches to demonstrate the innovative nature of this paper."

Reply: Many thanks to the reviewers for their suggestion: comparison with other methods. However, I would like to clarify because some of the inappropriate descriptions in the previous manuscript introduced the topic of this manuscript into fault diagnosis. In fact, this paper is mainly aimed at academics in the field of fault diagnosis by providing a platform for monitoring critical parameters in additive manufacturing processes. It is difficult to go in the direction of finding comparisons with than previous studies in this platform, so we apologise for the suggestions made by the reviewers. In addition, we have corrected the abstract, introduction, and conclusion sections of the manuscript in order to prevent our confusing descriptions from distracting readers from reading our manuscript.

How the paper is modified:   The before and after revisions about Abstract of our latest manuscript submission are as follows:

Before modification

After modification

The before and after revisions about introduction of our latest manuscript submission are as follows:

Before modification

After modification

The before and after revisions about conclusion of our latest manuscript submission are as follows:

Before modification

After modification

According to the reviewers’ comments, we have made extensive modifications to our manuscript and supplemented extra data to make our results convincing. Thank you again for your positive comments and valuable suggestions to improve the quality of our manuscript.

We appreciate for editors and reviewers’ warm work earnestly, and hope that the correction will meet with approval.

Yours sincerely,

Zhuoyong Shi

June 25, 2024

Reviewer 3 Report

Comments and Suggestions for Authors

The paper has been improved through revisions, and I believe it can be accepted.

Author Response

Many thanks to the reviewers for their excellent contributions to this paper!